# Enhanced Cytotoxic Effect of Doxorubicin Conjugated to Glutathione-Stabilized Gold Nanoparticles in Canine Osteosarcoma—In Vitro Studies

**DOI:** 10.3390/molecules26123487

**Published:** 2021-06-08

**Authors:** Anna Małek, Bartłomiej Taciak, Katarzyna Sobczak, Agnieszka Grzelak, Michał Wójcik, Józef Mieczkowski, Roman Lechowski, Katarzyna A. Zabielska-Koczywąs

**Affiliations:** 1Department of Small Animal Diseases and Clinic, Institute of Veterinary Medicine, Warsaw University of Life Sciences, Nowoursynowska 159c, 02-776 Warsaw, Poland; anna_malek@sggw.edu.pl (A.M.); roman_lechowski@sggw.edu.pl (R.L.); 2Department of Cancer Biology, Institute of Biology, Warsaw University of Life Sciences, Nowoursynowska 159, 02-776 Warsaw, Poland; bartlomiej_taciak1@sggw.edu.pl; 3Laboratory of Organic Nanomaterials and Biomolecules, Faculty of Chemistry, University of Warsaw, Pasteura 1, 02-093 Warsaw, Poland; katarzynasobczak@student.uw.edu.pl (K.S.); agrzelak@advantic.com.pl (A.G.); mwojcik@chem.uw.edu.pl (M.W.); mieczkow@chem.uw.edu.pl (J.M.)

**Keywords:** dogs, doxorubicin, flow cytometry, gold, humans, nanoparticles, osteosarcoma, P-glycoprotein

## Abstract

Osteosarcoma (OSA) is the most common malignant bone neoplasia in humans and dogs. In dogs, treatment consists of surgery in combination with chemotherapy (mostly carboplatin and/or doxorubicin (Dox)). Chemotherapy is often rendered ineffective by multidrug resistance. Previous studies have revealed that Dox conjugated with 4 nm glutathione-stabilized gold nanoparticles (Au-GSH-Dox) enhanced the anti-tumor activity and cytotoxicity of Dox in Dox-resistant feline fibrosarcoma cell lines exhibiting high P-glycoprotein (P-gp) activity. The present study investigated the influence of Au-GSH-Dox on the canine OSA cell line D17 and its relationship with P-gp activity. A human Dox-sensitive OSA cell line, U2OS, served as the negative control. Au-GSH-Dox, compared to free Dox, presented a greater cytotoxic effect on D17 (IC_50_ values for Au-GSH-Dox and Dox were 7.9 μg/mL and 15.2 μg/mL, respectively) but not on the U2OS cell line. All concentrations of Au-GSH (ranging from 10 to 1000 μg/mL) were non-toxic in both cell lines. Inhibition of the D17 cell line with 100 μM verapamil resulted in an increase in free Dox but not in intracellular Au-GSH-Dox. The results indicate that Au-GSH-Dox may act as an effective drug in canine OSA by bypassing P-gp.

## 1. Introduction

Osteosarcoma (OSA) is the most common primary bone neoplasia of mesenchymal origin, with an estimated incidence of 5.6–13.8/10,000 dogs [1,2,3,4]. It’s high metastatic potential is the main cause of death and poor prognosis. Metastasis mostly affects the lungs. While less than 15% of dogs have radiographically detectable metastases at diagnosis, it is believed that 80–90% of them have micrometastatic disease at that time [5,6]. The median survival time in dogs varies from 76 to 540 days, depending on the method of treatment (surgery, chemotherapy, or radiotherapy) [7,8]. Surgery, such as amputation or limb-sparing procedures with neoadjuvant and adjuvant chemotherapy, is considered the gold standard treatment for OSA. Nevertheless, the response of OSA to chemotherapy treatment varies between studies and, in the case of doxorubicin (Dox), is associated with potential adverse effects such as myelosuppression, cardiotoxicity, and gastrointestinal toxicity [9,10,11]. There are limited data on other investigational therapeutic agents for the management of metastasis, both in vitro and in dogs diagnosed with OSA; these include experiments on adjunctive bisphosphonate therapy with zoledronic acid [12,13,14], the use of liposomal muramyl dipeptide phosphatidylethanolamine [15], and phase I clinical trials on the role of immunotherapy with recombinant HER2-targeting *Listeria monocytogenes* [16] or interleukin 2 [17]. Radiation therapy in OSA is mostly used as adjunctive therapy in unresectable or incompletely resected tumors, as well as to reduce local bone pain and inflammation [18,19]. Another limitation of standard chemotherapy is related to multidrug resistance (MDR), mainly associated with high P-glycoprotein (P-gp) expression and activity. It prevents the intracellular accumulation of cytotoxic drugs (e.g., Dox, vinblastine, vincristine, and paclitaxel) in neoplastic cells, which is one of the main causes of chemotherapy failure [20,21]. The expression of proteins belonging to the ABC superfamily has been reported in various canine tumors, including pulmonary, mammary, hepatocellular carcinomas, and lymphomas [22]. In dogs with lymphoma receiving standard chemotherapy, higher *MDR1* expression is related to disease progression [23].

Liposomes or nanoparticles used as drug delivery systems are under extensive investigation to overcome the limitations of standard chemotherapy [24,25,26,27,28,29,30]. Gold nanoparticles (AuNPs) stabilized with a monolayer of l-aspartate increase the susceptibility of hepatic cancer cells (patient-derived cells and the hepatocellular carcinoma HepG2 cell line) to Dox, cisplatin, and capecitabine plus ribavirin, in comparison to cytostatic drugs alone [31]. Dox covalently conjugated to polyethylene glycol (PEG) PEGylated AuNPs, as opposed to free Dox, has been shown to achieve a higher concentration inside MDR HepG2-R cancer cells, probably due to the co-transportation of Dox bound to AuNPs and the bypass of P-gp. Efflux pump inhibition by AuNPs has been excluded as a mechanism of increased nano-drug accumulation in MDR HepG2-R cancer cells [32]. Cheng et al. showed enhanced cytotoxicity of folic acid-coated AuNPs conjugated to Dox, as well as improved drug accumulation and retention in comparison to free Dox in MDR HepG2-R cancer cells [33]. A study on ANS-TAT-AuNPs (anticancer molecule 2-(9-anthracenylmethylene)-hydrazine carbothioamide conjugated to AuNPs) showed an enhanced antiproliferative effect on HepG2 cells and a size-dependent effect on overcoming MDR in the MCF-7/ADR drug-resistant cell line [34]. ANS-TAT-AuNPs with diameters of 3.8 nm and 22.1 nm are both highly effective against MDR cells, although the larger 22.1 nm AuNPs show greater efficacy than the 3.8 nm AuNPs, probably due to P-gp size exclusion. Larger AuNPs are likely bigger than the P-gp drug efflux transporter, which could hinder drug efflux and more effectively overcome the MDR effect [33]. Moreover, Dox conjugated to glutathione stabilized gold nanoparticles (Au-GSH-Dox) previously demonstrated a high efficacy both in vitro and in ovo in Dox-resistant feline fibrosarcomas expressing high P-gp activity [35,36]. Therefore, the aim of this study was to investigate the influence of Au-GSH-Dox on a canine OSA cell line and its relation to efflux pump (mainly P-gp) activity.

## 2. Results

### 2.1. Au-GSH-Dox Treatment Increases Cytotoxicity in OSA Cell Lines

We sought to determine whether the non-covalent attachment of Au-GSH to Dox (Au-GSH-Dox) increases cell growth inhibition in comparison to Dox alone in canine and human osteosarcoma cell lines (D17 and U2OS, respectively), similar to what we previously showed for feline fibrosarcoma cell lines [35]. Interestingly, for the D17 cell line, the half-maximal inhibitory concentration (IC_50_) was 7.9 μg/mL and 15.2 μg/mL for Au-GSH-Dox and Dox, respectively (Figure 1A and Table 1). For the U2OS cell line, the IC_50_ was 8.9 μg/mL and 5.4 μg/mL for Au-GSH-Dox and Dox, respectively (Figure 1B and Table 1). The correlation between the concentration of the tested compounds and cell growth inhibition is presented in Figure 1. As suspected, based on the results of previously performed in vitro and in vivo studies [35,36,37], Au-GSH was non-toxic (cell viability > 80%) at each dose for both the D17 and U2OS cell lines (Appendix A). The results clearly indicate that the higher cytotoxic effect of Au-GSH-Dox in comparison to Dox alone was not due to Au-GSH cytotoxic activity. To further confirm the cytotoxic effect of the tested compounds (Au-GSH-Dox and Dox), as well as to assess the role of two forms of cell death (apoptosis and necrosis) induced by Dox and Au-GSH-Dox, annexin-V and propidium iodide (PI) dual staining was performed using IC_50_ concentrations.

### 2.2. Au-GSH-Dox Acts Mainly through Apoptotic Cell Death

As suspected for the D17 cell line, the number of apoptotic cells was significantly higher than necrotic cells (*** *p* ≤ 0.001) in the group treated with Au-GSH-Dox than in the group treated with Dox alone (IC_50_ dose of 7.9 μg/mL for Au-GSH-Dox) (Figure 2A). For the U2OS cells, statistically significant (*** *p* ≤ 0.001) differences between necrotic and apoptotic cells for both Au-GSH-Dox and Dox alone (IC_50_ dose of 5.4 μg/mL for Au-GSH-Dox) (Figure 2B) were observed, with similar percentages of cell death in both groups (Figure 2B and Figure 3B). Both Au-GSH-Dox and Dox act mainly via apoptotic cell death, which is in agreement with our previous studies on the Au-GSH-Dox and Dox mechanism of action in feline cancer cells [35]. No significant differences were observed in the number of apoptotic and necrotic cells between the control group and cells treated with Au-GSH alone in both the D17 and U2OS cell lines (Figure 2 and Figure 3), which also confirmed that Au-GSH is non-toxic. As presented in Figure 3, in early apoptosis, cells with intact membranes expose phosphatidylserine, which binds with annexin-V FITC. Cells in late apoptosis were labeled with annexin-V FITC and PI. PI binds to DNA in cells with damaged cell membranes. Necrotic cells were labeled with PI only (Figure 3).

### 2.3. Au-GSH-Dox Increases Cell Mortality in the D17 Cell Line

Furthermore, cell mortality was assessed using Trypan Blue staining. For the D17 cell line, after treatment with Au-GSH-Dox and Dox alone, at almost all tested concentrations, a statistically significant difference (*p* ≤ 0.01) was observed (Figure 4A)—as opposed to U2OS, where no difference was observed after treatment with Au-GSH-Dox and Dox alone, at almost all concentrations tested (Figure 4B). Treatment with Au-GSH did not cause a greater than 13% cell mortality in any concentration in D17 and U2OS, which was similar to the control (untreated) group (Appendix A).

### 2.4. Au-GSH-Dox but Not Au-GSH Alters OSA Cell Morphology

Apoptosis is characterized by a series of typical features involved with cell morphology changes such as cell shrinkage [38], which we observed after treatment with the tested compounds. Then we compared the morpho-physiological features of the D17 and U2OS cell lines. The D17 cells had a spindle shape, longer protrusions, and were larger than the U2OS cells, whereas the U2OS cells were more compact than the D17 cells, with a spindle-to-triangle-shaped morphology. For both the D17 and U2OS cell lines, no floating cells or cells with altered morphology were visible under the microscope in either the control group or after treatment with Au-GSH, at all tested concentrations, which further verified our statement and that of other scientists that Au-GSH alone is non-toxic. A higher number of floating cells, greater cell shrinkage, and changes in morphology (shorter protrusions, a round shape, and a smaller size), as well as cytoplasmic contraction, were observed in the groups treated with increasing concentrations of Au-GSH-Dox and Dox for both tested cell lines (Figure 5 and Appendix A), which corresponds to the morphological hallmarks of apoptosis.

### 2.5. P-gp Inhibition with Verapamil Enhances the Intracellular Accumulation of Rhodamine 123 in the D17 Cell Line

Rhodamine 123 is a fluorescent dye broadly used to assess P-gp activity, which acts as a substrate for P-gp-related extracellular efflux in MDR cells [39,40]. Verapamil is a potent P-gp inhibitor widely used as a modulator of P-gp activity in MDR cell lines [41,42,43]. The cytotoxic effect of Au-GSH-Dox on the D17 and U2OS cell lines may positively correlate with an increased MDR phenotype, similar to the results from previously published studies, indicating the enhanced cytotoxicity of Au-GSH-Dox in feline fibrosarcoma cell lines resistant to Dox [35]. Indeed, for the D17 cell line, slight differences in the fluorescence intensity were observed between cells treated with rhodamine 123 alone (a P-gp fluorescence substrate) and cells with an additional 20 μM verapamil (a P-gp modulator) (Figure 6A). This is different from the U2OS Dox-sensitive OSA cell line lacking P-gp activity, for which no difference was observed in rhodamine 123 accumulation after the addition of 20 μM verapamil (Figure 6B).

### 2.6. P-gp Inhibition with Verapamil Increases the Intracellular Accumulation of Free Dox but Not Au-GSH-Dox in the D17 Cell Line

The fluorescence intensity of the D17 cells increased after treatment with Dox and 100 μM verapamil in comparison to cells incubated with Dox alone. This indicates that free Dox was actively pumped out of the cells by efflux pumps and blocking them by verapamil (P-gp modulator), increasing the intracellular Dox concentration. D17 cells treated with Au-GSH-Dox alone and Au-GSH-Dox with 100 μM verapamil had similar fluorescence intensities in both groups. The results obtained indicate that the intracellular concentration of Au-GSH-Dox is not related to P-gp inhibition with verapamil, which was clearly evident for the D17 cell line. This is in agreement with a study from Gu and collaborators [32], indicating that efflux pump inhibition by AuNPs is excluded as a mechanism of increased nano-drug accumulation in MDR HepG2-R cancer cells. For U2OS, there was no difference in the fluorescence intensity in the groups treated with Dox alone and Dox with the addition of 100 μM verapamil. This is consistent with previously obtained results with rhodamine 123 and verapamil tests. Both tests clearly presented a lack of greater accumulation of either Dox or rhodamine 123 after verapamil efflux pump inhibition, which corresponds with the U2OS doxorubicin-sensitivity and the lack of P-gp activity. Similarly, the same results (no difference in fluorescence) were obtained for U2OS cells treated with Au-GSH-Dox with and without 100 μM verapamil (Figure 7). The results obtained are in agreement with a previous study on feline fibrosarcomas [35], indicating the positive correlation between the increased cytotoxicity of Au-GSH-Dox and cell lines with the MDR phenotype.

## 3. Discussion

Nanoparticles and liposomes are used as drug delivery systems for cytostatic drugs to increase the intracellular accumulation of the drug in target cells and to reduce the negative side effects of standard chemotherapeutics [44,45,46]. In human medicine, several nanoparticle-based drugs are registered by the Food and Drug Administration, and others are currently undergoing clinical trials [47]. In veterinary medicine, there are just a few preclinical studies on the use of liposomes and nanoparticles to improve cancer treatment in dogs and cats [48,49]. A few clinical trials of liposomal Dox have been performed on dogs to test their use in humans [26,50]. For canine OSA, Higginbotham and collaborators [51] showed a decreased viability of D17 cells after 96 h of treatment with micellar nanoparticles containing Dox, in comparison to cells treated with Dox alone, with the lowest effective dose being 1 μM. Moreover, carbon magnetic nanoparticles (40–50 nm) conjugated with Dox, starting from a dose of 15 μg/mL, caused a dose-dependent inhibition of D17 cell proliferation, indicating their possible use in targeted drug delivery systems [52]. As cell-based assays are needed to determine the cytotoxic effects of compounds [53] investigated for potential therapeutic application, this was the first step of the present study.

It was shown that Au-GSH alone was non-toxic at all tested concentrations (ranging from 10 to 1000 μg/mL) for both cell lines (Appendix A), which is similar to previously published studies on its use in feline fibrosarcomas [35,36]. AuNPs have been revealed as non-toxic in many other studies, both in vitro and in vivo. AuNPs-GSH with an average diameter of approximately 2.1 ± 0.3 nm showed zero to low (<20%) cytotoxic activity on human OSA cells (143B), human osteoblasts (hFOB1.19), breast cancer cells (MCF7), breast epithelial cells (MCF10A), pancreatic cancer cells (PANC-1), and pancreatic cells (hTERT-HPNE cells) [54]. In addition, the previously mentioned AuNPs complexes did not induce cytotoxic effects, such as in cells treated with 10 nm of nanocomposite with porphyrin-embedded AuNPs (TPPS-AuNPs) up to concentrations of 200 μM for 72 h [55] and ANS-TAT-AuNPs (3.8 nm and 22.1 nm) with concentrations up to 100 µg/mL, incubated for 24 h [34]. Moreover, similarly to our studies, empty AuNPs showed no toxicity up to 200 μM in A549 (a small cell lung cancer line) and WI-38 (normal lung fibroblasts) [55]. Furthermore, upon intraperitoneal administration of colloidal AuNPs (12.5 ± 1.7 nm) in doses of 40, 200, and 400 μg/kg/day for eight consecutive days in mice caused no mortality and no gross changes in survival, behavior, animal weight, organ morphology, blood biochemistry, or tissue histology [56]. More importantly, because of the similarity in the formulation and size of the AuNPs used in the present study, glutathione-coated AuNPs (1.2 ± 0.9 nm) caused no morbidity at concentrations up to 60 μM in a murine model [57].

In the present study, Au-GSH-Dox was more cytotoxic than free Dox for the D17 cell line (with P-gp activity visible by the rhodamine 123 and verapamil tests) per the results of the MTT assay (IC_50_ was 7.9 μg/mL and 15.2 μg/mL for Au-GSH-Dox and Dox, respectively; Table 1). This was confirmed in cell mortality and apoptosis assays (Figure 2 and Figure 4), which indicates that the conjugation of Dox to Au-GSH facilitated an almost two-times lower dose of Dox to induce cell death mainly through apoptosis in the D17 cells. These results agree with previously published in vitro studies on the application of Au-GSH-Dox for feline fibrosarcomas with high P-gp activity [35]. In the future, the enhanced cytotoxic effect of Au-GSH-Dox in in the D17 cell line could contribute to a possible decrease in the adverse effects of chemotherapy due to a significantly lower dose of the cytostatic drug. There was no significant difference in the cell mortality and induced apoptosis in the U2OS cell line treated with Au-GSH-Dox and Dox alone (Figure 2 and Figure 4). The U2OS cell line is known to lack P-gp, multidrug resistance-associated protein 1 (MRP1/ABCC1), and breast cancer resistance protein (BCRP/ABCG2) expression, leading to Dox sensitivity [58]. This further confirms the hypothesis that Au-GSH-Dox works more effectively in MDR cells. This finding is in agreement with a study on human OSA where dextran nanoparticles (112.4 ± 4.2 nm; +1.19 ± 0.82 mV) loaded with Dox were shown to overcome MDR, as increased application facilitated drug accumulation inside Dox-resistant KHOS_R2_ and U2OS_R2_ cells, which showed increased apoptosis compared to free Dox. The fluorescence intensity of resistant cells treated with nanoparticles was comparable to that of drug-sensitive ones (KHOS and U2OS) [30]. In the present study, U2OS displayed no difference in fluorescence intensity in either rhodamine 123 and Dox accumulation assays, with or without verapamil treatment, which is consistent with previous Dox sensitivity studies [58].

Mechanisms related to MDR and ABC transporters are well-known in humans, which have 48 ABC transporters. Three of them typically associated with drug resistance are P-gp (MDR1/ABCB1), MRP1/ABCC1, and BCRP/ABCG2 [59]. Only a few studies have assessed P-gp expression and function in canine OSA. High P-gp expression has been found in an OSA cell line treated with Dox [60] and in a sporadic appendicular canine OSA sample [61]. The high activity of efflux pumps correlates with a low intracellular concentration of rhodamine 123 due to active outflow [42]. Verapamil is a well-known blocker of P-gp [41]. An indirect assessment of P-gp activity is possible by the evaluation of the fluorescent intensity of cells treated with rhodamine 123 and a P-gp inhibitor that prevents outflow and increases the concentration of rhodamine 123 inside the cells [62]. Previously, high BMI1 expression, which contributes to tumor cell growth and chemotherapy resistance, was described for D17 cells [63]. Treatment with PTC-209 (BMI1 inhibitor) in combination with 30 nM Dox was shown not to cause a significant decrease in cell viability, which indicates that different mechanisms play a key role in D17 Dox-resistance. In the present study, we determined P-gp activity in the D17 cell line, which revealed a greater rhodamine 123 accumulation after P-gp inhibition with 20 μM verapamil. The drug accumulation assay further confirmed a slight increase in Dox accumulation in the D17 cells after verapamil treatment in comparison to cells incubated with Dox alone. Mealey et al. found that canine OSA cells resistant to Dox (OS2.4/doxo) accumulate significantly less Dox than sensitive ones (OS2.4). Moreover, OS2.4/doxo cells treated with Dox and verapamil show higher fluorescence intensities than when treated with Dox alone, which is consistent with our findings [60].

Importantly, inhibition of the D17 cell line with 100 μM verapamil results in an increase in free Dox, but not in intracellular Au-GSH-Dox, indicating that the higher cytotoxic effect of Au-GSH-Dox is not related to higher levels of P-gp fluorescence substrates (rhodamine 123 as well as Dox) after inhibition with verapamil. Similar results for AuNPs were observed by Gu and collaborators [32] for MDR HepG2-R cell line. The authors claimed that the preferential uptake of AuNPs observed in MDR HepG2-R cells is probably due to an endocytic entry, bypassing P-gp, and a connection with the fluidity of the cell membrane in MDR cells [32]. In a study with both chitosan (CMC)-based AuNPs loaded with Dox (Dox/CMC-AuNPs) and PEGylated Dox-loaded CMC-based AuNPs (Dox/CMC-AuNPs-PEG), the efflux ratio in the Caco-2 cell line was reduced in comparison with Dox, which may indicate that both CMC-AuNP complexes with Dox overcome P-gp-mediated MDR [64]. In the LN229 cell line (MDR glioblastoma multiforme), a Dox-loaded nanocomposite with porphyrin-embedded AuNPs (DOX@TPPS-AuNPs) has been shown to increase cellular uptake and significantly reduce drug efflux in comparison to free Dox [55]. The effect of an increased intracellular accumulation was observed in the previously mentioned study with ANS-TAT-AuNPs [34]. In this case, accumulation was believed to be involved with their AuNPs that surpassed the size of the P-gp channel, which prevented drug efflux. Additionally, PEG-modified AuNps with succinimidyl 4-(Nmaleimidomethyl) cyclohexane-1-carboxylate (SMCC) and conjugated with Dox (Au-SMCC-DOX) showed an increased accumulation inside MDR HepG2-R cells in comparison to free Dox [33]. Finally, the results of the present study are consistent with previously published studies testing Au-GSH-Dox in feline fibrosarcomas [35,65], where the cytotoxic effect was positively correlated with their P-gp activity.

## 4. Materials and Methods

### 4.1. AU-GSH-Dox and Au-GSH

Au-GSH (4.3 ± 1.1 nm) and the complex of Au-GSH-Dox (4.3 ± 1.2 nm) were synthesized according to a previously described procedure [35,65], determined using transmission electron microscopy and dynamic light scattering [35,65]. The zeta potential of the particles was equal to −52.9 ± 10.6 mV and −24.6 ± 11.9 mV for Au-GSH and Au-GSH-Dox, respectively, which confirmed the stability of Dox binding to Au-GSH [36].

All reagents were purchased from Sigma Aldrich (St. Louis, MO, USA) unless otherwise stated.

### 4.2. Cell Culture

This study utilized the D17 (ATCC, Manassas, VA, USA) canine OSA cell line and the U2OS (Sigma Aldrich, St. Louis, MO, USA) human OSA cell line. D17 is a commercially available cell line derived from an OSA metastasis to the lung of an 11-year-old female poodle. U2OS is a commercially Dox-sensitive human OSA cell line with low P-gp expression and activity [30,66,67], derived from a moderately differentiated sarcoma of the tibia of a 15-year-old girl. The D17 and U2OS cell lines were cultivated in Eagle’s Minimum Essential Medium (EMEM) and McCoy’s 5A Modified Medium with 2 mM L-glutamine, respectively, with the addition of 10% v/v heat-inactivated fetal bovine serum (FBS), penicillin-streptomycin (50 mL IU-1), and amphotericin B (2.5 mg/mL). The cells were maintained under standard conditions (5% of CO_2_, 95% humidity, and 37 °C). The media was changed every 48–72 h, and the cells were split when cell confluence reached 70–80%.

### 4.3. MTT Assay

The thiazolyl blue tetrazolium bromide (MTT) assay is widely used to evaluate cytotoxicity and cell viability, as the amount of formazan is directly proportional to the number of living cells [68]. The MTT assay was performed to determine the cytotoxic effect of Au-GSH-Dox, Au-GSH, and Dox alone. The D17 and U2OS cells were seeded in 96-well plates (Becton Dickinson, USA) at a concentration of 3 × 10^4^ cells per well. After 24 h of incubation, the media containing tested compounds at 11 various concentrations were added: 0.1, 0.25, 0.5, 1, 2.5, 5, 7.5, 10, 25, 50, and 100 μg/mL for free Dox and Dox in Au-GSH-Dox and 2, 5, 10, 20, 50 100, 150, 200, 500, 1000, and 2000 μg/mL for Au-GSH (as the Dox/Au-GSH concentration ratio in Au-GSH-Dox was 1:20). Medium without tested substances was added to the control group. The cells were incubated for 24 h. After 24 h, the media were removed, and 0.5 mg/mL of tetrazolium salt was added to each well for 4 h. To complete solubilization of the formazan crystals, 100 μL of dimethyl sulfoxide (DMSO) was added to each well. Photometric absorbance was measured at 570 nm using the multi-well plate reader Infinite 200 PRO Tecan (TECAN, Mannedorf, Switzerland).

### 4.4. Annexin-V and Propidium Iodide Dual Staining

Analysis of the apoptosis in OSA cells was performed by flow cytometry using annexin-V FITC and PI dual staining. The D17 and U2OS cells were seeded in 6-well plates and incubated until they reached 70–80% confluency. In the next step, the media were replaced with medium containing tested compounds and the calculated IC_50_ concentrations for Au-GSH-Dox. The cells were harvested after 24 h of incubation, washed in cold phosphate-buffered saline (PBS), transferred to 5 mL round-bottom polystyrene test tubes (Becton Dickinson, Franklin Lakes, NJ, USA), and stained with an Annexin V Kit (Becton Dickinson, Franklin Lakes, NJ, USA), following the manufacturer’s protocol. The OSA cells were analyzed by flow cytometry (BD FACS Aria II, Becton Dickinson, Franklin Lakes, NJ, USA) and separated from cellular debris, and doublets of FSC/SSC and FSC-A/FSC-H gates were analyzed with detection filters: 530/30 (annexin-V FITC) and 610/20 (PI), using an excitation laser with a wavelength of 488 nm.

### 4.5. Cell Mortality

First, 5 × 10^5^ cells (D17 and U2OS) per well were seeded in a 6-well plate. After 24 h of standard incubation, the media were removed, and the tested compounds were introduced to the cells at increasing concentrations: 5, 10, 20, and 50 μg/mL for Dox and Dox in Au-GSH-Dox and 100, 200, 400, and 1000 μg/mL for Au-GSH. No substances were added to the medium of the control group. The cells were incubated with the tested substances for 24 h, then harvested by trypsinization and centrifuged at 1800 rpm for 3 min. After this, 10 μL of the cell suspension was mixed with 10 μL of 0.4% Trypan Blue and cell counts were evaluated with a Countess II FL Automatic Cell Counter (Thermo Fisher Scientific, Waltham, MA, USA). Dead cells were stained blue, and cell mortality was expressed as the percentage of dead cells (from the total cell number).

### 4.6. Cell Morphology

The D17 and U2OS cells were plated in 6-well plates (5 × 10^5^ cells per well) and incubated for 24 h. The media were replaced with medium containing examined compounds at the increasing concentrations mentioned above. The substance concentrations were chosen based on the results of the cell mortality and MTT assays, selecting concentrations that resulted in mostly 20–80% of cell death in both cell lines and included IC_50_ doses of Dox and Au-GSH-Dox. No substances were added to the media of the control group. Twenty-four hours later, the cell morphology was assessed with an inverted optical microscope (Primo Vert, Zeiss, Oberkochen, Germany).4.7. Test with Rhodamine 123 and Verapamil.

The D17 and U2OS cells were first harvested by trypsinization and transferred into 5 mL round-bottom polystyrene test tubes (Becton Dickinson, Franklin Lakes, NJ, USA) at 1 × 10^6^ cells per tube. The cells were divided into three groups: Medium only as a negative control, 1 mM rhodamine 123 as a positive control, and 1 mM rhodamine 123 with 20 μM verapamil. Cells were incubated at 37 °C for 1 h. After 1 h, the cells were washed twice with cold PBS, analyzed by flow cytometry (BD FACS Aria II, Becton Dickinson, Franklin Lakes, NJ, USA), separated from cellular debris, and doublets were measured using FSC/SSC and FSC-A/FSC-H gates and analyzed with a 530/30 filter detecting rhodamine 123, using the excitation laser at a 488 nm wavelength.

### 4.7. Au-GSH-Dox and Dox Accumulation

The D17 and U2OS cells were seeded into 6-well plates (Becton Dickinson, Franklin Lakes, NJ, USA) at a concentration of 4 × 10^5^ per well. After a 24 h incubation, the tested cells were treated with 100 μM verapamil for 1 h in serum-free medium. Then, Dox or Au-GSH-Dox was added to the media at the IC_50_ concentration and incubated for 1 h. The control cells were treated with Dox and Au-GSH-Dox only. At this time, the cells were harvested, washed in PBS, and analyzed by flow cytometry (BD FACS Aria II, Becton Dickinson, Franklin Lakes, NJ, USA). Analysis was performed by separating cellular debris from the doublets using FSC/SSC and FSC-A/FSC-H gates, the 585/42 detection filter, and the excitation laser at a wavelength of 488 nm. All of the experiments were performed in triplicate.

### 4.8. Statistical Analysis

The statistical analysis was conducted using the Student’s *t-*test for the mortality assay, and a one-way ANOVA and post-hoc Tukey’s test for the annexin-V and PI tests and IC_50_ determination from the dose-response curve using GraphPad Prism 5.0 (San Diego, CA, USA). A *p*-value ≤ 0.05 (*) was determined as significant, while *p* ≤ 0.01 (**) and *p* ≤ 0.001 (***) as highly significant.

## 5. Conclusions

Significantly higher cytotoxic effects of Au-GSH-Dox, in comparison to free Dox, along with P-gp activity was determined for the D17 cell line. AuNPs were non-toxic at all of the tested concentrations (ranging from 10 to 1000 μg/mL). After verapamil treatment, the greater intracellular accumulation of Dox, but not Au-GSH-Dox, indicates that the Au-GSH-Dox mechanism of action may be due to bypassing the P-gp transporter, and Au-GSH-Dox may act as an effective drug in Dox-resistant OSA. Further molecular and in vivo studies are needed to confirm its efficacy and to assess the molecular mechanism of action.

## Figures and Tables

**Figure 1 molecules-26-03487-f001:**
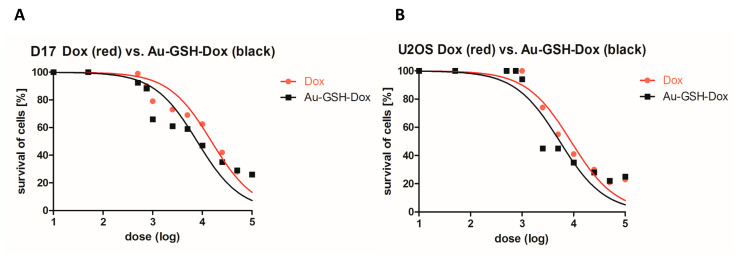
Correlation between the logarithm of Dox (red line) and Au-GSH-Dox (black line) doses and cell viability (measured by MTT assay) for D17 (**A**) and U2OS (**B**) cell lines.

**Figure 2 molecules-26-03487-f002:**
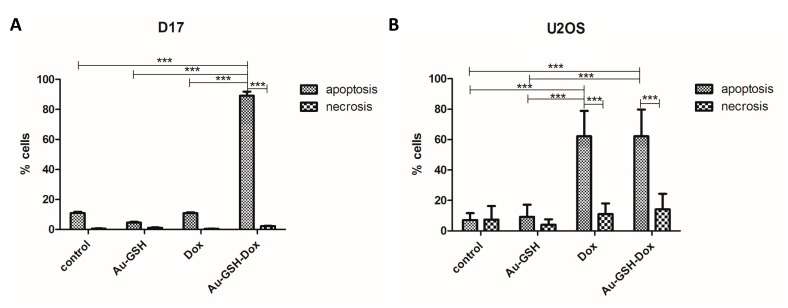
Statistical analyses of the percentage of cells in apoptosis and necrosis measured with annexin-V and PI assay on D17 (**A**) and U2OS (**B**) cell lines treated with Au-GSH-Dox and Dox in an IC_50_ Au-GSH-Dox dose measured with the MTT assay. *** *p* ≤ 0.001 was interpreted as highly significant.

**Figure 3 molecules-26-03487-f003:**
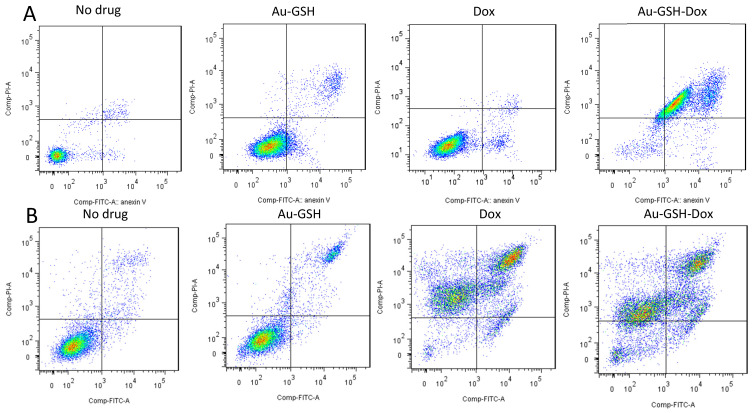
Scatter diagram of D17 (**A**) and U2OS (**B**) cells treated with tested substances in annexin-V and PI assay, measured by flow cytometry (treated with Au-GSH-Dox and Dox in an IC_50_ Au-GSH-Dox dose measured by MTT assay).

**Figure 4 molecules-26-03487-f004:**
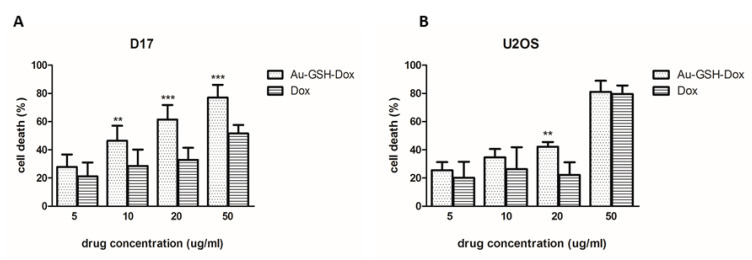
Effect of the tested substances (Au-GSH-Dox and Dox) on the mortality of the tested cell lines: D17 (**A**) and U2OS (**B**). ** *p*
*≤* 0.01 and *** *p* ≤ 0.001 were interpreted as highly significant.

**Figure 5 molecules-26-03487-f005:**
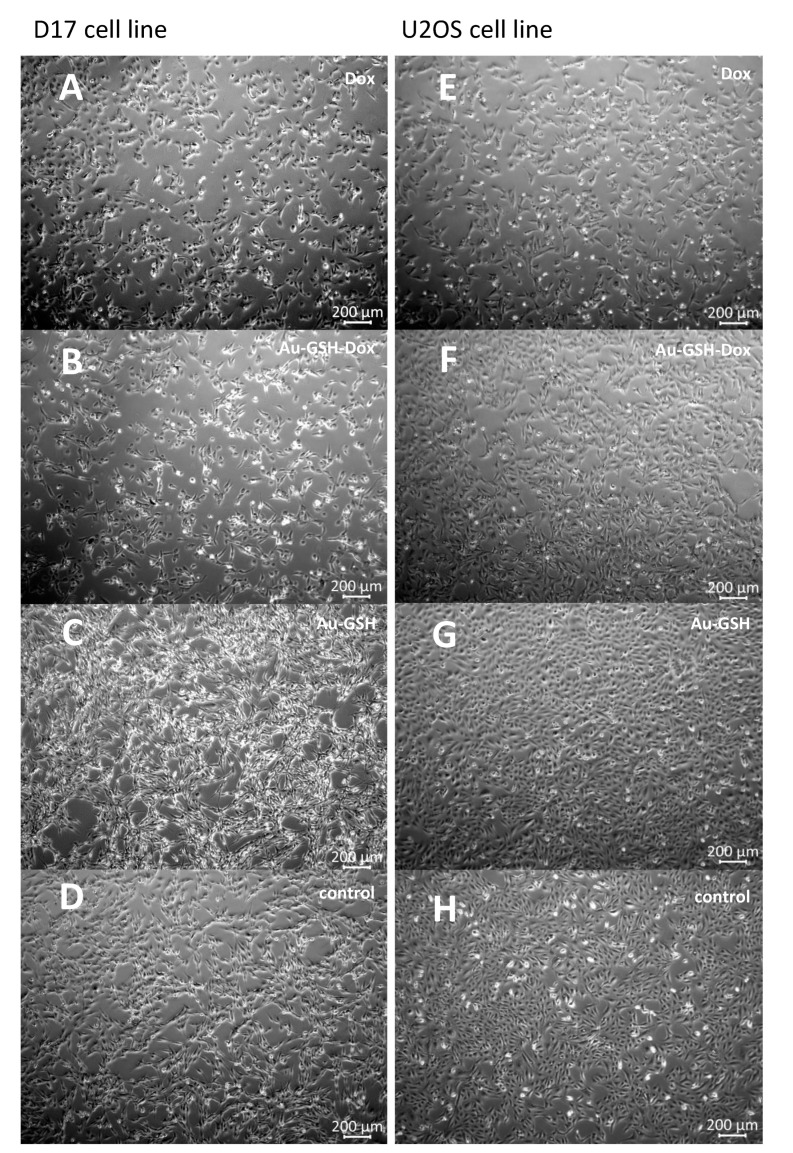
Phase-contrast microscopy images of D17 and U2OS OSA cells treated with 5 μg/mL of Dox (**A**,**E**), 5 μg/mL of Au-GSH-Dox (**B**,**F**), 100 μg/mL of Au-GSH (as the Dox/Au-GSH concentration ratio in Au-GSH-Dox was 1:20) (**C**,**G**), and untreated (negative control) (**D**,**H**). 4× magnification, scale bar 200 μm.

**Figure 6 molecules-26-03487-f006:**
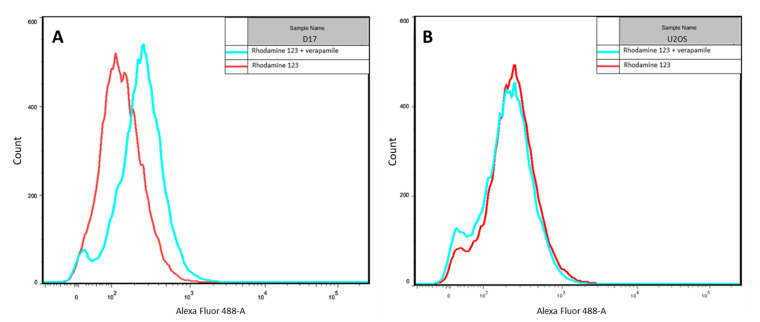
Fluorescence of rhodamine 123 with or without verapamil in treated D17 (**A**) and U2OS (**B**) cell lines.

**Figure 7 molecules-26-03487-f007:**
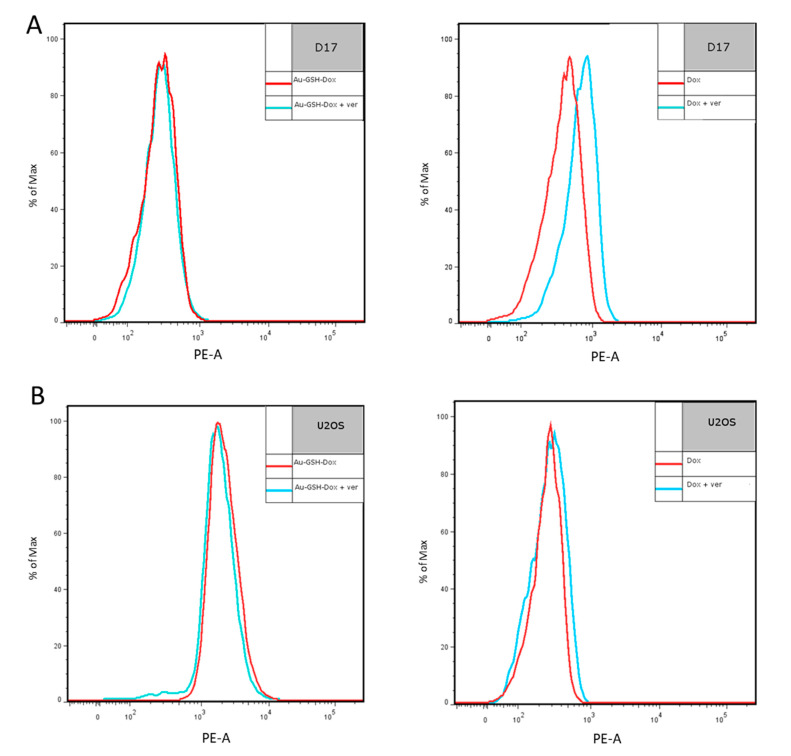
Fluorescence of D17(A) and U2OS (B) after treatment with Au-GSH-Dox or Dox alone with or without verapamil, analyzed by flow cytometry.

**Table 1 molecules-26-03487-t001:** Half-maximal inhibitory concentration (IC_50_) of Dox and Au-GSH-Dox for the D17 and U2OS cell lines.

Cell Line	IC_50_ Dox (μg/mL)	IC_50_ Au-GSH-Dox (μg/mL)
D17	15.2	7.9
U2OS	8.9	5.4

## Data Availability

All data generated and analyzed during this study are included in this published article.

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
