# Peer review of "Enhanced Cytotoxic Effect of Doxorubicin Conjugated to Glutathione-Stabilized Gold Nanoparticles in Canine Osteosarcoma—In Vitro Studies"

_molecules, 2021, doi:10.3390/molecules26123487_

Round 1

Reviewer 1 Report

Major revision

This reviewer  suggests an exhaustive revision of the language and style of the manuscript.

This reviewer considers that some points, particularly in terms of description of the results, also need to be addressed. In the results sections, the organization reflects the techniques/assay used and not the biological results and its relation with the action of the compounds. This reviewer suggest that this should be improved, and that an explanation of the purpose of the different assays should be included for context, but not represent the main part of the description of results.

In methodological terms, the authors compare a canine osteosarcoma cell line with an human cell line. Is there available a canine cell line without the MDR phenotype for comparison? This is probably more relevant than the comparison of 2 very different cell lines. 

The Figures need to be improved:

Figure 1 – change colour of size bar

Figure 3  - Graphs should be uniform – same axis- The curve fitting does not seem the best – p.ex for Augsh dox in D17 and U2OS. 

Some examples of points to address in the text 

Introduction

5,6-13,8/10,000 in dogs    - please clarify – is it 5.6 to 13.8  in 10 000 animals?

Line 57  -   MDR1 gene expression but  MDR abbreviation – only in line 65

Line 60  - Overcome limitations- what limitations?

Line 72  - Size- dependent effect? Please explain or remove

Line 74  - Rephrase

Reviewer 2 Report

This paper is about the influence of AU-GSH-Dox on D17 canine osteosarcoma cell line which expresses P-gp activity comparing with U2OS an human doxuribicin-sensitive osteosarcoma cell line. The paper follow an abstract introduction, results, discussion, material and methods, conclusions, references and adequate supplementary materials that helped the interpretation of the results and discussion. All the components are well written but there are some details that the authors must have in account.

Introduction section, line 34: the estimated incidence should be confirmed if the values are 5.6-13.8/10,000 instead what the text show.

Results section: the paragraph identification should be corrected:

Line 94: 2.2 instead 2.1

Line 104: 2.3 instead 2.2

Line 116: 2.4 instead 2.3

Line 138: 2.5 instead 2.4

Line 145: 2.6 instead 2.5

Line 262: 4.2 instead 4..2

Line 263: ATTC should appear ATCC

Line 103: caption of the figure 2: the graph of the figure 2 shows ** and *** - what means each?

The caption should express the meaning of these signs.

The graphs of the figure 3 represents the percentage of cell survivor should present the same scale for the both graphs. We recommend that the value of 50% should be highlighted.

Figure 4: in spite of the authors explain in the text the comparison done it is not clear in the figure 4. The authors should clarify what is the comparison with control and the individual comparison.

Abbreviation section: there are several abbreviations that does not appear in the section (eg. KHOS, MRP1, BMI1, BCRP, …). Once the authors decide to have abbreviation list, all of them should be shown.

References section: some of the references does not have DOI. The authors should harmonize according to the journal rules.

Reviewer 3 Report

The paper of Malek and colleagues describes the possibilities fo Au-GSH-Dox to act as a promising drug in osteosarcoma treatment bypassing the P-gp.

The paper is of highly significant especially because it evidences the possibility to use doxorubicn in osteosarcoma treatment avoiding the commonly known drug resistence effects. The authors obtained appreciable results.

I have minor concerns regarding some specifications that should, in my opinion, be added to the paper.  

  • the reasons for which Dox and Au-GSH-Dox doses (5, 10, 20, 50 µg/mL) were established should be precisely explained within the text. If some preliminary data regarding the choice of these doses exist, they have to be added to the paper.
  • RESULTS SECTION- cell morphology-  Figure 1 of supplementary material ad figure 1 of the text describes images as "optical microscopy images" this should be replaced by "phase contrast images" and magnification shoud be reported; furthermore, when authors describe figure 1 of supplementary materials, they provide a lot of cell details which are not appreciable with this low magnification, thus, I strongly recommend to add some higher magnification images  supplemented with arrows or lines to indicate described details.
  • RESULTS SECTION- apoptosis assay- in figure 4A and 4B the statistical analysis needs to be better explained in the figure legends by adding to which samples significance refers to, "significant" or "highly significant" is not sufficient if samples are not indicated.
  • MATERIALS AND METHODS- cytotoxicity and viability assay- This information is not correct as MTT test only evaluates cell metabolic activity without providing information regarding cytotoxicity. Cytotoxic parameters can be evaulated with different analyses such as Lactate Dehydrogenase test. Thus, if authors need to provide information regarding cytoxicity they should perform more appropriate analyses, if not the title of paragraph 4.5 has to be corrected deleting "cytotoxicity". 
  • MATERIALS AND METHODS- apoptosis assay- replace this title with a more specific one such as "Annexin V-PI dual staining".

typing or spelling mistakes:

-introduction, line 75 "in vivo" instead of " in ovo";

  • materials and methods, line 105 "tested compounds" instead of "test compounds".

English revision performed by a native speaker is also recommended.

Round 2

Reviewer 1 Report

This reviewer appreciates the changes introduced by the authors, which have made the manuscript easier to read and understand.

However, in this reviewer´s opinion some modifications are still needed in order to improve the overall understanding and presentation of the relevant work developed by the authors.

Moreover, an extensive revision by a English native speaker with a science background is still required.

Please find below a list of modifications that this reviewer considers relevant:

Introduction

Lines 74- 77

“ANS-TAT-AuNPs with diameters of 3.8 nm and 22.1 nm were greatly effective against MDR cells, whereas larger AuNPs (22.1 nm) had higher efficacy probably due to the effect of P-gp size-exclusion. The size of P-gp blocks the entry of conjugated AuNPs into the cell [33].”

I appreciate the effort for clarification. However, the sentence is somewhat confusing….

3.8nm and 22.1 nm are effective, and 22.1 more so than 3.8 – is that it?

Also the conclusion is not correct – “The size of P-gp blocks the entry of conjugated AuNPs into the cell”

The original article refers in the abstract: “ The data suggest that the larger AuNPs had more profound effect on overcoming MDR, which could effectively prevent drug efflux due to their size being much larger than that of the p-glycoprotein channel”

So 2 points to be clarified:

It is the drug  efflux, not entry,  that is affected and it is the size of the Aunps that causes the difference and not the size of Pgp (that does not change)

Results

Lines 83-84

A brief explanation f the 2 cells lines should be introduced here in the beginning of the result section

Also the new Supl Figure 1 is wrongly identified in the legend as Supl figure 3, please change

Line 143

Initial sentence seems strange… I suggest linking it with the second sentence…

Line 162

2.5 Rhodamine 123 and verapamil slightly affect fluorescence intensity.

Here the authors did not reformulate the presentation of results as they did in the previous sub-sections.

I must suggest that they change this and integrate it better with the following studies with the Aunps.  The results in Figure 6 must be presented as a way to prove the MDR phenotype of the 2 cell lines. The authors describe the results, but they must explain why using rhodamine (fluorescent substrate of pgp) and verapamil ( pgp modulator), this is never mentioned.

In line 164, please replace the word assumed… the authors wanted to test if, similarly to previous results, in the canine osteosarcoma, the cytotoxicity could be correlated with a MDR phenotype. Assumptions are not relevant here.

Then the results from Figure 7 are easier to present and explain, but again please mention that the experiments were made, taking advantage of the fluorescence of dox.

In line 179 : “ .. higher P-substrate levels after inhibition…. “ please explain: higher levels of Au.gsh-dox? Higher levels of PgP? It is not clear.  And this sentence is also in the discussion (line 248) so correction there is also necessary.

Discussion

Line  201 – Au-GSH-Dox was cytotoxic than free Dox - … please rephrase – more cytotoxic than…

Line 206 – apoptosis and mortality assays evaluate cell death… not cytostatic effect.

In the discussion the authors first explore the MDR phenotype and in the final part discuss the effect of the Aunps alone. I suggest moving this part of the Aunps to the beginning of the discussion and then focus on the Au-gsh-dox versus dox alone and the relation with the mdr phenotype. The papers mentioned in the introduction (31 to 36).  need to be included in the discussion, as they help to explain the potential way of escaping MDR.

Conclusions

Line 372

Please rephrase “achieved higher dox… “ In the present form it does not make sense.
